# Intima–Media Thickening with Carotid Webs: A Case Report of a Potentially High-Risk Association

**DOI:** 10.3390/diagnostics15212756

**Published:** 2025-10-30

**Authors:** Corrado Tagliati, Alessia Quaranta, Marco Fogante, Claudio Ventura, Stefania Lamja, Alfonso Alberto Matarrese, Pierpaolo Palumbo, Iacopo Carbone, Ernesto Di Cesare, Gabriele Polonara, Nicolò Schicchi

**Affiliations:** 1AST Ancona, Ospedale di Comunità Maria Montessori di Chiaravalle, Via Fratelli Rosselli 176, 60033 Chiaravalle, Italy; 2AST Macerata, Cardiologia, Distretto Sanitario di Civitanova Marche, Via Abruzzo, 62012 Civitanova Marche, Italy; alessiaquaranta84@gmail.com; 3Maternal-Child, Senological, Cardiological Radiology and Outpatient Ultrasound, Department of Radiological Sciences, University Hospital of Marche, Via Conca 71, 60126 Ancona, Italy; marco.fogante89@gmail.com (M.F.); claudioventura20@gmail.com (C.V.); nicolo.schicchi@ospedaliriuniti.marche.it (N.S.); 4Department of Biotechnological and Applied Clinical Sciences, University of L’Aquila, Via Vetoio, 67100 L’Aquila, Italy; stefanialamja@gmail.com (S.L.); palumbopierpaolo89@gmail.com (P.P.); ernesto.dicesare@univaq.it (E.D.C.); 5AST Ascoli Piceno, Cardiologia, Ospedale Mazzoni, Via degli Iris 1, 63100 Ascoli Piceno, Italy; alfonsomatarrese@gmail.com; 6Department of Radiological, Oncological and Pathological Sciences, Academic Diagnostic Imaging Division, I.C.O.T. Hospital, Sapienza University of Rome, Via F. Faggiana 1668, 04100 Latina, Italy; iacopo.carbone@uniroma1.it; 7Department of Specialized Clinical Sciences and Odontostomatology, Polytechnic University of Marche, 60126 Ancona, Italy; g.polonara@staff.univpm.it

**Keywords:** carotid web, intima–media thickening, intima–media thickness, stroke, high risk, Doppler ultrasound, Spectral Doppler waveform, turbulent flow, carotid web length, web-to-bulb ratio

## Abstract

We describe a case of an asymptomatic 70-year-old female patient on whom a carotid ultrasound examination was performed that showed intima–media thickening and a 4 mm long carotid web with a 50% web-to-bulb ratio. Spectral Doppler waveform demonstrated a turbulent flow pattern and a peak systolic velocity increase of 100% (velocity ratio = 2) when compared with the common carotid artery. Therefore, the patient seemed to be at risk of stroke, and antiaggregant treatment was suggested.

**Figure 1 diagnostics-15-02756-f001:**
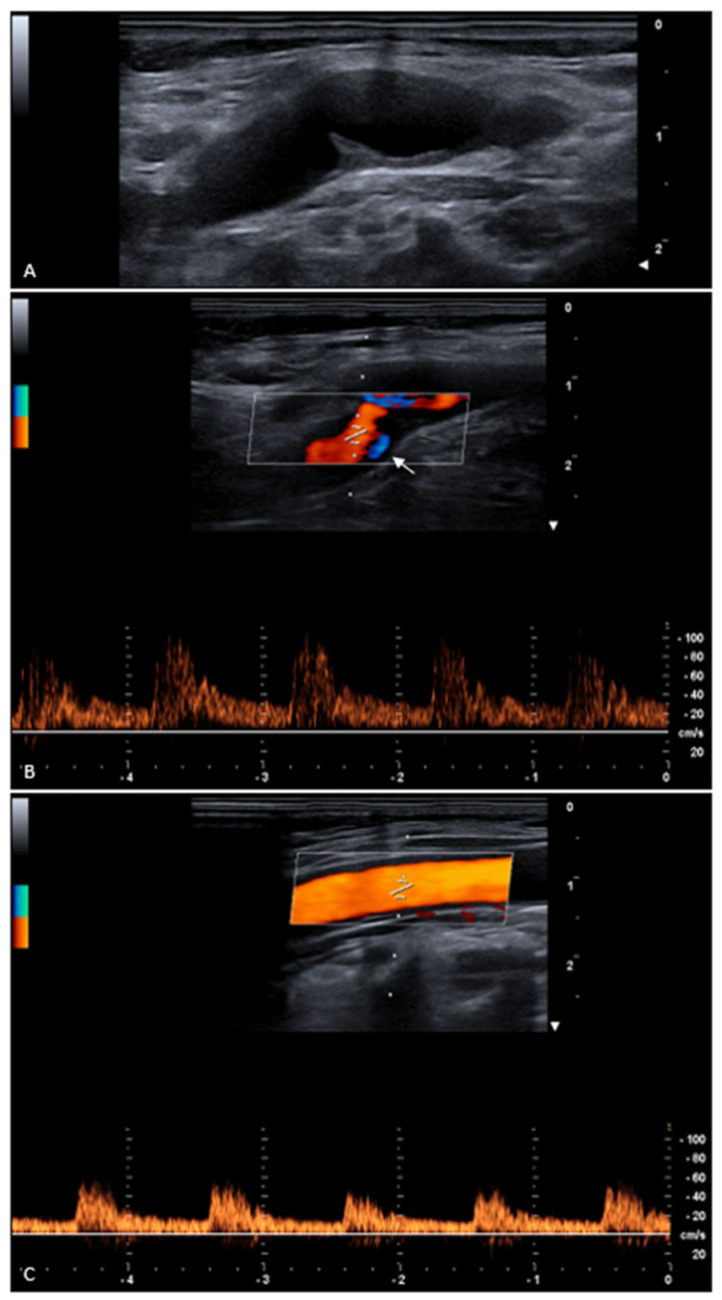
Long-axis carotid B-mode ultrasound showed a 1.3 mm intima–media thickening and a 4 mm long triangular-shaped intraluminal filling defect in the posterolateral wall of the proximal portion of the right internal carotid artery, which is a carotid web (**A**). It determined a reduction in the internal carotid lumen, with a 50% web-to-bulb ratio, and turbulent flow was generated at the angle between the carotid web and the internal carotid artery wall on the Doppler ultrasound image (**B**, arrow). Spectral Doppler waveform demonstrated both peak systolic velocity and an end-diastolic velocity increase of 100% (velocity ratio = 2) when compared with the right common carotid artery (**C**). Therefore, the patient seemed to be at risk of stroke, and antiplatelet treatment was suggested, particularly aspirin-based single antiplatelet therapy. A combination therapy of ezetimibe 10 mg plus atorvastatin 20 mg was recently started due to hypercholesterolemia, and we confirmed it, although its benefit remains speculative with regard to the carotid web treatment. Moreover, abdominal and peripheral arterial Doppler ultrasound examinations were recommended. After six months of follow-up, neither transient ischemic attack nor stroke occurred, and a neurologist visit confirmed current treatment, with the suggestion to carry out close ultrasound and clinical checks.

A carotid web is a triangular luminal irregularity usually arising from the posterolateral wall of the proximal internal carotid artery near carotid bifurcation, and it causes larger local hemodynamic disruption and larger regions of thrombogenic flow stasis than mild and moderate atherosclerotic plaques [1,2,3]. A carotid web is not usually found in pediatric strokes, and its forming process is not clearly understood, but it is considered to be a focal intimal variant of fibromuscular dysplasia [4].In the past, a carotid web was thought to be a rare condition. However, nowadays, it is known that about 0.5–1% of people have a carotid web, and the prevalence of bilateral carotid webs is not so rare as it could be up to about 0.05 [5,6,7]. Moreover, it is possible that a portion of carotid webs are not diagnosed, and their prevalence could be even higher.A previous article suggested to rename the carotid web as the cervical artery web, as other arteries where rarely found to be affected, such as subclavian and vertebral arteries [8].A longer carotid web length and a higher degree of web stenosis were statistically associated with stroke [9,10,11]. An in-depth analysis of carotid web angioarchitecture showed a statistically significant association between angular measurements and stroke status, in particular a common carotid artery-web-pouch angle of ≥41.7°, an internal carotid artery web-pouch angle of ≥92.4, a common carotid artery-pouch-tip angle of ≥89.4°, and an internal carotid artery pouch-tip angle of ≥85.7° [12,13] (Appendix A).In our patient, the carotid web length was 4 mm, the degree of stenosis was 50%, the common carotid artery web-pouch angle was 67°, and the common carotid artery pouch-tip angle was 93°, which are potential morphological high-risk features.Intima–media thickening is not a normal finding, even in older patients. It is true that intima–media thickness increases with age, but at the age 70 it is considered normal up to 0.75 mm. Therefore, 1.3 mm is definitely abnormal, as in our patient. Intima–media thickening could cause an increase in carotid web length, as carotid web is an intimal lesion. Therefore, as statin treatment can reduce the increase in intima–media thickening and reduce the risk of plaque development, statins could be justified in this setting [14].We performed B-mode and Doppler ultrasound to diagnose the carotid web, and these techniques are recognized as valuable tools for carotid web diagnosis, particularly the longitudinal view [1,15,16]. Microvascular imaging can be useful for carotid web detection, particularly for very thin ones [14]. Contrast-enhanced ultrasound can help detect atherosclerotic plaques and thrombosis associated with carotid webs [17]. However, other previously published studies suggested that computed tomography and magnetic resonance imaging could detect carotid webs better [18]. Digital subtraction angiography has been reported by some as the gold-standard imaging modality for carotid web detection, and it can show contrast stagnation and its duration, but it is an invasive technique not recommended for diagnosis [18,19]. However, the lack of knowledge could lead to misdiagnosis and missed diagnosis with all diagnostic techniques [20].Not as many carotid webs have been described in the literature with hemodynamic effects. In fact, in a case series of 24 patients with carotid webs, no one showed those effects [15]. Another study reported that, in 94% of patients, the carotid stenosis was less than 50% [21]. Moreover, a previously published article reported that 7 out of 68 carotid webs showed at least 50% stenosis [22].Our patient did not show a carotid plaque, but carotid webs with associated atherosclerotic plaques were previously reported [20,22]. It was described that the atherosclerotic plaque can be placed in two different locations, under the carotid web or parallel to it [20]. The same study reported that a higher patient age is significantly associated with carotid webs and concomitant atherosclerotic plaques. Moreover, atherosclerotic plaque presence significantly raises the probability of a stenosis to higher than 50% [20], and, in general terms, plaque geometric features such as surface irregularities and the length of upstream segment are associated with plaque vulnerability and the risk of stroke [23]. Furthermore, atherosclerotic plaque can raise the probability of carotid web misdiagnosis [24]. However, to the best of our knowledge, this is the first case of carotid web with abnormal intima–media thickness, and no previous articles have discussed about potential high-risk carotid webs associated with intima–media thickening.The differential diagnoses of carotid webs are atherosclerotic plaques, particularly atherosclerosis plaque surface ulceration, carotid dissection, and intraluminal thrombus; however, carotid webs and these diseases can be concomitant [7,25,26].Atypical carotid webs, characterized by unusual location, abnormal morphology, or association with concurrent diseases, such as intraluminal thrombus, dissection, or atherosclerotic plaques, can make carotid webs particularly challenging to diagnose, leading to potential misdiagnosis and missed diagnosis [26,27,28]. A case report showed that Doppler ultrasound revealed a carotid web with a lodged thrombus, whereas digital subtraction angiography and computed tomography were not able to diagnose it [28]. Sometimes, only subsequent examinations can reveal that a carotid web with a lodged thrombus was at the base of carotid stenosis [29]. At times, in symptomatic patients, the definitive diagnoses of carotid webs can be reached only after carotid endoarterectomy or after the re-evaluation of previously acquired images [27,28,30]. However, a case report showed that, in a patient with an acute vestibular syndrome and an incidental diagnosis of carotid webs, the latter was not present in a computed tomography angiogram examination performed seven years earlier [31]. Therefore, radiologists and clinicians cannot always trust a previous examination with normal carotid arteries to exclude a carotid web in a symptomatic patient with atypical carotid findings.Carotid web treatment is suggested in symptomatic carotid webs after a stroke or a transient ischemic attack, where carotid endarterectomy or stenting are the preferred treatments to prevent future ischemic events [17,32,33]. However, our patient was asymptomatic, and no consensus guidelines exist about asymptomatic carotid web treatment; therefore, future studies need to determine what is the course of action in these not so rare patients [34,35]. Is antiplatelet treatment recommended in all asymptomatic patients with a carotid web? Is antiplatelet treatment enough in asymptomatic patients with carotid webs with high-risk features? How long does the follow-up need to be after carotid web diagnosis to be sure that the patient is really asymptomatic?In conclusion, we describe a case of an asymptomatic 70-year-old female patient on whom a carotid ultrasound examination was performed that showed intima–media thickening and a carotid web with high-risk features; therefore, antiplatelet treatment was suggested to reduce the risk of possible cerebrovascular accidents. Future research and guidelines need to guide clinicians on how to manage asymptomatic patients with carotid webs.

## Data Availability

Data are contained within the article.

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
