# Peer review of "Intima–Media Thickening with Carotid Webs: A Case Report of a Potentially High-Risk Association"

_diagnostics, 2025, doi:10.3390/diagnostics15212756_

Round 1

Reviewer 1 Report (Previous Reviewer 2)

Comments and Suggestions for Authors

Thank you for this submission and the opportunity to

perform a review. The authors describe the case of a

70-year-old asymptomatic lady who was incidentally

diagnosed as having a carotid web and accompanying

carotid intimal-medial thickening, which is a sign of

atherosclerosis. The images are of high quality,

however, the association of a carotid web with carotid

atherosclerosis seems a very long shot since almost

anyone at the age of 70 years will much probably have

a thickened intima. I do not think that this report carries

much educational value to the readers.

Author Response

Thank you very much for your comments.

We strongly agree that the association of a carotid web with carotid plaques was previously reported in some articles. However, 1.3 intima thickening is a high degree thickening. In fact, it is true that intima-media thickness raises with age, but at the age 70 it is expected to be normal up to 0.75 mm. Therefore, 1.3 mm is definitely abnormal, as in our patients. Moreover, to the best of our knowledge, this is the first case of carotid web with abnormal inti-ma-media thickness and no previous articles discussed about potential high-risk carotid web associated with intima-media thickening arotid web  

Many thanks.

Reviewer 2 Report (New Reviewer)

Comments and Suggestions for Authors

The observation of carotid plaques plays a key role in early detection and prevention of stroke. Overall, the paper is well written. I suggest to extend the discussion on the diversity of carotid plaques in component, morphology, geometry, and hemodynamic effect (e.g., 10.1016/j.heliyon.2024.e37419).

Author Response

Thank you very much for your comments.

The discussion was extended adding a sentence about plaque geometry, as suggested.

Manny thanks

Reviewer 3 Report (New Reviewer)

Comments and Suggestions for Authors

This is a very interesting case report, however I have few comments:

-The association between intima-media thickening and carotid web as a possible high-risk condition is interesting but not fully developed. The authors should expand the discussion on why this association could increase cerebrovascular risk, beyond age-related thickening, and whether this modifies diagnostic or therapeutic strategies.

-Please clarify if this is the first reported case of carotid web with abnormal intima-media thickness, as suggested, and provide a stronger rationale supported by literature.

-The patient was started on aspirin and continued on statin/ezetimibe therapy. Given the uncertain role of medical therapy in carotid web, the rationale for this choice should be more critically discussed. What evidence supports aspirin in asymptomatic carotid webs with high-risk features? Could a closer neurological evaluation or earlier intervention have been considered?

-Finally, the discussion reviews several papers, but it reads somewhat like a list rather than a synthesis. A more critical comparison between prior studies and the present case would highlight the manuscript’s contribution more clearly.

Author Response

Thank you very much for your comments.

1. Intima-media thickening could cause a raising of carotid web length, as carotid web is an intimal lesion. Therefore, as statin treatment can reduce the increase in intima-media thickening and reduce the risk of plaque development, statins could be justified in this setting. Hese sentence was added.

2. This is the first reported case of carotid web with abnormal intima-media thickness. This sentence was added.

3. The neurologic visit was performed and the treatment was confirmed. “and a neurologist visit confirmed current treatment, with the suggestion to carry out close ultrasound and clinical checks.” was added.

4. Some sentences were added to allow a more critical comparison between prior studies and the present case “We performed B-mode and Doppler ultrasound to diagnose the carotid web, and these techniques are recognized as valuable tools for carotid web diagnosis, particularly the longitudinal “, as in our patient” “Our patient did not show a carotid plaque, but” “However, our patient was asymptomatic, and no consensus guidelines exist about asymptomatic carotid web treatment; therefore”.

Many thanks.

Round 2

Reviewer 2 Report (New Reviewer)

Comments and Suggestions for Authors

Thanks for the update. My previous comments have been addressed.

Reviewer 3 Report (New Reviewer)

Comments and Suggestions for Authors

Thank you for the revisions.

This manuscript is a resubmission of an earlier submission. The following is a list of the peer review reports and author responses from that submission.

Round 1

Reviewer 1 Report

Comments and Suggestions for Authors

Case report of a 70-year-old, asymptomatic patient diagnosed with a carotid-WEB lesion.
Well-written article, extensive discussion, but I have a few questions.

Comments:
1/ Since the authors mentioned the morphology of the lesion, which is associated with an increased risk, please provide a schematic drawing of the angles and dimensions of the lesion that are associated with an increased risk of neurological complications.
2/ What antiplatelet therapy do the authors consider optimal (or perhaps anticoagulant therapy) - SAPT (ASA, clopidogrel, ticagrelol), DAPT?
3/ Do statins play a protective role, and is there evidence to support their use in these patients - which and dose?
4/ What is the optimal duration of antiplatelet and anticoagulant therapy? Are there ultrasound parameters indicating a low risk of complications, and does the patient then require antiplatelet or anticoagulant therapy?
5/ what is the authors' opinion on the procedure - a young patient with a carotid WEB type lesion in angioCT, but no lesion visible in ultrasound - what next - control angioCT, DSA+IVUS examinations?

Author Response

Thank you very much for your suggestions and comments.

  1. A schematic drawing of how are measured the angles that are associated with an increased risk of neurological complications is attached as a supplementary material.
  2. In a patient without previous antiplatelet treatment without previous TIA/stroke we consider sufficient SAPT (ASA). No consensus exists in literature about the best thing to do in asymptomatic patients.
  3. Statins can avoid an increase of intima-media thickening and avoid a plaque formation; in fact, if a plaque arises near or over a carotid web it could increase the total volume of the lesion and increase the risk of complications due to higher flow turbulence and higher stenosis. Therefore, we think statins can play a role in reducing this risk. Our patient has recently started a combination therapy (ezetimibe 10 mg plus atorvastatin 20 mg) and we did not suggest to modify it.
  4. There is no consensus in literature about asymptomatic patients. We think that if an antiplatelet treatment is started it needs to be continued forever. If a low risk of complication is suggested by ultrasound parameters in our opinion the suggestion to start antiplatelet therapy is less strong, but the treatment could be started depending on physician and patient’s choice.
  5. I think carotid web are visible on a well performed ultrasound examination at the bifurcation and in the first tract of the internal carotid artery (ICA). Obviously, in rarer cases in which the web is detected in angioCT in the distal portions of ICA, CT could see a web and US not as the distal portion is not evaluable with US. In the latter case, angioCT could be repeated. However, if the web is at the bifurcation, in the first tract of ICA or in other US evaluable vessels, an US needs to be repeated by an expert practitioner and in exceptional cases contrast-enhanced ultrasound could be performed.

Many thanks

Reviewer 2 Report

Comments and Suggestions for Authors

Thank you for this submission and the opportunity to perform a review. The authors describe the case of a 70-year-old asymptomatic lady who was incidentally diagnosed as having a carotid web and accompanying carotid intimal-medial thickening, which is a sign of atherosclerosis. The images are of high quality, however, the association of a carotid web with carotid atherosclerosis seems a very long shot since almost anyone at the age of 70 years will much probably have a thickened intima. I do not think that this report carries much educational value to the readers. 

Author Response

Thank you very much for your comments.

To the best of our knowledge, few previously published articles described atherosclerotic plaques associated with carotid webs.

No previously published articles described carotid intimal-medial thickening associated with carotid webs. Moreover, no previously published articles described carotid intimal-medial thickening associated with carotid webs with high-risk features.

Many thanks.

Reviewer 3 Report

Comments and Suggestions for Authors

I read with interest the Interesting Images article entitled “Intima-media thickening and carotid web: a high-risk association?” The authors have presented clear and valuable images of carotid web and intima-media thickening, which represent findings that are rarely encountered and sometimes overlooked in clinical practice. While the manuscript is clinically relevant, I have several comments:

The number of co-authors appears disproportionate to the scope and length of this type of manuscript. A more concise authorship list reflecting substantial contributions would be more appropriate.

The title is formulated as a question; however, carotid web is already well recognized in the literature as a high-risk condition for ischemic stroke. The use of a question mark may therefore be misleading, and a clearer title without it is recommended.

Including a computed tomography (CT) image of the presented case would enhance the quality and completeness of the manuscript.

The statement “antiplatelet treatment was suggested” should be specified—please clarify which antiplatelet drug was prescribed.

It would be useful to indicate which clinical department or team is following the patient.

The discussion section is generally informative; however, there are repeated phrases such as “A previous article,” “Previous studies,” and “A previously published article.” Since it is implicit that cited work is previously published, these repetitions are unnecessary. Revising the writing to avoid redundancy would improve readability.

Author Response

1.Thank you very much for this suggestion. However, the case is exceptional, and a lot of authors evaluated the case, a lot of authors were needed to decide if writing this manuscript was useful, a lot of authors were necessary to find and read existing literature, a lot of physicians were needed to decide what to suggest to the patient, a lot of authors prepared the original draft, reviewed and edited it. Therefore, how can I ask to some authors who substantially contributed to the manuscript, and that read and agreed to the published version of the manuscript to accept to be removed from it? I thank you very much for your suggestion; I surely will ask help to less physicians in the future to try to have a shorter author list. However, at the moment it is difficult to remove somebody who substantially contributed to it.

2.The title was changed into: “Does intima-media thickening increase the risk of carotid web complications?”

3.The patient did not perform a CT examination, as in our opinion the diagnosis is evident, and CT examination was not suggested. We think that radiation exposure is not justified at the moment, taking into account ICRP (publication 103).

4.PT (ASA) treatment was suggested.

5.A neurologist is going to follow the case. The patient booked an appointment that she will have in the future: the waiting lists are long.

6.Some “previous article” were removed.

Many thanks

Round 2

Reviewer 3 Report

Comments and Suggestions for Authors

The authors have provided the necessary responses, and I thank them for their efforts. Of course, it is not possible to request the removal of contributing authors, and your point about the multidisciplinary approach is well taken. However, given the large number of co-authors, one would reasonably expect that such a strong collaboration could also lead to the preparation of a high-quality review article in this field.

Author Response

Thank you for your comments and suggestions.

1.The title was modified as expected

2.CT was not performed and I cannot oblige to perform an examination only for publication purposes. At the moment, ultrasound is enough in the author’s opinion. If in the future a vascular surgeon will request CT it will be performed.

  1. Intima-media thickening is not a normal finding, in older patients too. It is true that intima-media thickness raises with age, but at the age 70 it is expected to be normal up to 0.75 mm. Therefore, 1.3 mm is definitely abnormal, as in our patients. These sentences were added to the manuscript.
  2. Spectral Doppler waveform demonstrated both peak systolic velocity and end-diastolic velocity increase of 100% (velocity ratio = 2) compared with the right common carotid artery. This sentence was modified as requested.
  3. A statement was added, as requested. “The authors take full responsibility of the case report presented”.
  4. The authors know that combination therapy with atorvastatin/ezetimibe evidence for web-specific benefit remains speculative, and this was acknowledged as such.

7.”Previously published studies/article” were reduced.

  1. I spoke with the authors, and some accepted to be removed in order to avoid problems with the Journal.

9.Language was reevaluated.

Many thanks.